# Nutrition-related knowledge, attitudes, practice, and mental health status of people living with HIV in Dubai, United Arab Emirates: A cross-sectional descriptive study

**Souheir M. Alia[1], Taoufik L. Zoubeidi[2,¤], Habiba I. Ali[1]***

**1** Department of Nutrition and Health, College of Medicine and Health Sciences, United Arab Emirates University, Al Ain, United Arab Emirates, **2** Department of Analytics in the Digital Era, College of Business and Economics, United Arab Emirates University, Al Ain, United Arab Emirates

¤ Interdisciplinary School of Health Sciences, Faculty of Health Sciences, University of Ottawa, Canada
* habali@uaeu.ac.ae

## Abstract

### Background

Despite significant advancements in management, Human Immunodeficiency Virus (HIV) infection remains a global health concern. Limited research exists on the nutrition-related knowledge, attitudes, and practices (KAP) of people living with HIV (PLHIV) in the Middle East. This study assessed nutrition-related KAP and mental health status among PLHIV attending an HIV management clinic in Dubai, United Arab Emirates (UAE).

### Methods

A cross-sectional study was used to assess HIV/AIDS nutrition-related KAP. The total KAP scores ranged from 13–69, with scores between 13–32 representing poor, 33–51 average, and 52–69 good KAP. The Hospital Anxiety and Depression Scale (HADS) was used to screen the patients for anxiety and depression. Data was collected face-to-face from August to November 2023.

### Results

Data were collected from 63 patients attending an outpatient clinic. The mean age of the participants was $40 \pm 12.4$ years, and 80% were men. Most participants (87.3%) demonstrated good KAP (84.6%% and 89.8% for females and males, respectively). Mental health screening for depression and anxiety showed mean total scores of $4.73 \pm 3.6$ and $4.9 \pm 4.9$, respectively, which correspond to normal levels. The prevalence of abnormal HADS depression and HADS Anxiety were 8.0% and 17.%, respectively.

**Data availability statement:** The data supporting the findings of this study are available within the paper and its Supporting information. Due to ethical considerations, the authors cannot make any additional interview data publicly available because they contain potentially identifiable and confidential patient information, with the possibility that individuals may be recognized in the data. Additional data may be shared upon reasonable request to Mohammed Bin Rashed University Research Ethics Committee; email: irb@dubaihealth.ae.

**Funding:** The author(s) received no specific funding for this work.

**Competing interests:** The authors have declared that no competing interests exist.

## Conclusions

This study found that the participants had good nutrition-related KAP scores, with the prevalence of depression and anxiety less than 20% which is significantly lower than estimates for PLHIV worldwide. This is the first study to explore KAP and mental health status among PLHIV in the Arabian Gulf region.

## 1. Introduction

Human Immunodeficiency Virus (HIV) damages the immune system and can cause AIDS if left untreated. HIV has killed 36.3 million people [1]. With better HIV prevention, diagnosis, treatment, and care, HIV has become a manageable chronic health condition that allows HIV-positive people to live long and healthy lives [2].

In 2010–2023, new HIV infections in the Middle East and North Africa (MENA) grew 116% [3]. The HIV response is far from 2025 coverage targets. If governments adequately serve at-risk persons, the area may reduce new HIV infections despite low prevalence. HIV disproportionately affects inmates and outcasts [3] and is mainly due to transmissions through sex partners and drug use. Almost 20% of new HIV infections in the region were in 15–24-year-olds, primarily male (55%). These epidemic trends emphasize the need for scaled-up HIV interventions for key populations, particularly adolescents, and eliminating social and structural obstacles to crucial services.

Building on this, the UAE continues to have a very low HIV prevalence, according to the WHO country profile, and the epidemic is categorized as low-level in the 2020 profile [4]. Although some subsequent estimates place the prevalence among those aged 15–49 at approximately 0.3% in 2023 [5], nationally disaggregated data on the overall number of HIV-positive individuals are not made public [5]. Moreover, the mental health of people living with HIV is at stake, since the stress of living with a serious illness like HIV can damage mental health. HIV-positive people are more likely to develop mood, anxiety, and cognitive impairments [6]. HIV-positive patients often experience depression which in turn can be an added reason to their unhealty or risky behaviors [6].

When it comes to HIV, people's risky behaviors like having multiple sexual partners, having unprotected intimacy, using drugs via needles, poor dietary habits and not getting periodically tested all contribute to the higher prevalence of the HIV epidemic [7,8]. On the contrary, good habits like sound nutritional habits, getting tested regularly, compliance with treatment, harm reduction methods when injecting drugs, and healthy behaviors decrease its prevalence [7,8].

Poor nutritional awareness leads to poor diets, while good nutritional knowledge promotes healthy eating [9]. HIV/AIDS patients' nutritional knowledge and behaviors vary between countries [10]. A previous study examined nutritional KAP in adult HIV/AIDS patients at an AIDS Outpatient clinic in Nigeria and found BMI and ART duration were significantly correlated with good KAP [11]. Another study found a significant relationship between knowledge and energy intake but no significant relationship between attitude and food intake [12].

Poor nutritional awareness leads to poor diets, while good nutritional knowledge promotes healthy eating [9]. HIV/AIDS patients' nutritional knowledge and behaviors vary between countries [10]. A previous study examined nutritional KAP in adult HIV/AIDS patients at an AIDS Outpatient clinic in Nigeria and found Body Mass Index (BMI) and anti-retroviral therapy (ART) duration were significantly correlated with good KAP [11]. Another study found a significant relationship between knowledge and energy intake but no significant relationship between attitude and food intake [12]The stress of living with a serious illness like HIV can damage mental health. HIV-positive people are more likely to develop mood, anxiety, and cognitive impairments [6]. HIV-positive patients often experience depression [6].

Several studies have conducted surveys on public knowledge and attitudes toward PLHIV in the MENA region, including Saudi Arabia [13], UAE [14], Oman [15], Qatar [16], Yemen [17], Jordan [18], Lebanon [19], and Syria [20]. A systematic review and meta-analysis found that 74.4% of health workers had good HIV/AIDS knowledge, but only 52.8% had positive attitudes [21]. These results showed that despite ample knowledge, views remained mostly negative, requiring more efforts to construct educational programs to destigmatize PLHIV.

Studies involving PLHIV in the MENA region are limited. Moreover, to our knowledge, no studies have been conducted with PLHIV in the Arabian Middle East region. Thus, this study examined the nutrition knowledge, attitudes, and practices (KAP – primary outcome), as well as mental health status (depression and anxiety – secondary outcome) of people living with HIV (PLHIV) attending an outpatient clinic in Dubai, United Arab Emirates.

The rationale behind the secondary outcome is that psychological well-being is a known determinant of HIV-related knowledge, attitudes, and health practices; mental health status was included as a secondary outcome. PLHIV are often affected by depression, anxiety, and stigma, which can affect how people view health information, participate in lifestyle interventions, and follow advised practices [22]. Reduced motivation, low self-efficacy, poor health decision-making, and suboptimal ART adherence are all closely linked to poor mental health, and these factors may limit or moderate improvements in KAP outcomes [23]. Evaluating mental health is consistent with holistic HIV care frameworks that prioritize integrated mental health support and allow for a more thorough interpretation of behavioral outcomes [24]. A recent meta-analysis indicates that roughly 35% of individuals living with HIV (PLHIV) globally exhibit depressive symptoms (95% CI: 31–38%) [25]. However, a separate systematic review indicated that the aggregated prevalence of anxiety disorders among individuals living with HIV was roughly 15.5% [26]. Limited data indicate greater burdens in the MENA region; in North African sub-regions, depression prevalence among PLHIV has been reported to reach 41% [27].

## 2. Methods

### 2.1. Study design

This cross-sectional descriptive study evaluated the nutrition-related KAP and mental health status (depression and anxiety) of people living with HIV (PLHIV) in the UAE before their enrollment in an interventional study. Participants were recruited from among patients attending a major outpatient follow-up clinic for HIV in Dubai, UAE.

Based on hospital statistics, at the time the study was conducted, an average of 20–25 patients visited the outpatient HIV clinic every week, with a total of approximately 175 patients attending the clinic, of whom 20% were females. A random sample stratified by sex (20% females and 80% males) was used to select participants. The inclusion criteria were patients >18 years of age, those with the virus with or without any comorbidities, and those who consented to participate. Patients who were imprisoned, refused to participate, or <18 years of age were excluded.

### 2.2 Sample size calculations

The sample size was determined to achieve an adequate estimation precision for the mean total score of the primary variable KAP. Estimating the mean total KAP score in a finite population of 175 patients, with 95% confidence and an estimation error of 1.4, assuming a SD of 7, requires a sample size of 63. The value of the SD was inferred from the study

of Baghel et al. [28]. The selected sample corresponds to 36% of the total population of HIV patients attending the clinic at the time of the study.

## 2.3 Data collection

Cross sectional data were collected from the study participants face-to-face at the time of recruitment. All the 63 patients who met the eligibility criteria and consented to participate in the study provided data through an interviewer-administered questionnaire developed from relevant literature, including nutrition-related KAP questions for HIV patients at an Indian outpatient clinic [28]. The questionnaire measured all three components of KAP, assigning scores of 0–15 for knowledge, 15–45 for attitude, and 0–15 for practice. The overall score of the original questionnaire ranged from 15 to 75, with 15–35 representing "poor" KAP values, 36–55 "average", and 56–75 "good." The questionnaire underwent face and content validity assessments, which modified the KAP score ranges to 0–14 for knowledge, 14–42 for attitude, and 0–13 for practice. The total KAP score and category ranges were also modified to be 13-69 for the total score, where 13-32 represent "poor," 33-51 "average," and 52-69 "good" total KAP scores.

The Hospital Anxiety & Depression Scale (HADS) was used to screen the mental health status of the study participants [29]. Patients used a validated Arabic version of HADS [30]. A high HADS score above 11 indicates anxiety and depression, requiring additional evaluation by a psychologist and maybe a psychiatrist [31]. Patients' medical records provided weight, height, and BMI. The data collection period was from 28 August 2023–28 November 2023.

## 2.4 Validating of KAP questionnaire

The questionnaire was derived from the literature [28] and initially constructed in English. It was face and content validated, back-translated into Arabic, and professionally proofread. Five health professionals assessed the questionnaire's face validity, and 10 panelists (five dietitians, three Infectious Disease Unit physicians, one HIV clinical coordinator, and one infectious disease unit nurse) used Critical Values for Lawshe's Content Validity Ratio (CVR) to assess its content validity [32,33]. The score was 0.92, indicating good question validity. The questionnaire was administered in Arabic since all participants were native Arabic speakers during their regular visits to the clinic.

The questionnaire was revised and pilot-tested with 10 HIV-positive outpatients from the HIV clinic. The questionnaire was revised based on the feedback of the participants and experts. Finally, a competent translator translated and proofread all KAP subsections of the study questionnaire from English to Arabic. A validated and reliable Arabic version of the Hospital Anxiety and Depression Scale was available from the literature [30]. The questionnaire consisted of 41 questions, which required 6-10 minutes to complete it.

## 2.5 Ethics approval

The Research Ethics Committee at Mohammed Bin Rashed University and the Dubai Students' Research Ethics Committee approved the protocol (MBRU IRB-2023–72). All participants were informed that their participation in the study was voluntary and that their data would be kept private. Each participant gave verbal and written consent before data collection. The lead investigator had sole access to all study data to ensure confidentiality and oversaw all operational aspects of the research [34].

## 2.6. Data analysis

Data were analyzed using IBM SPSS Statistics (28.0). Descriptive statistics are presented as frequencies, proportions, means, and standard deviations. Group comparisons were performed using Kruskal-Wallis test, and Analysis of Variance. Given the very high data completion rates, with minimal item-level missing data in few variables, and the absence of systematic patterns in missing values, analyses were conducted using complete cases without imputation. The statistical test used for each analysis is specified in the results tables. Statistical significance was determined at p-value <0.05.

## 3. Results

All study participants were UAE nationals; moreover, the mean age of the participants was 40 years (SD = 12.4) (Table 1). Most of the participants were either single (47.6%) or married (38.1%). Only 17.5% had no income, whereas most earned over 15,000 UAE dirhams/month. In terms of education, 47.6% had a high school education, whereas only 6.3% held a graduate degree.

The various nutritional supplements used by the participants are presented in S1 Fig. Only 35% of the study participants consumed oral nutritional supplements, of which 50% consumed a nutritionally medicated fluid supplement, 22.7% consumed a multivitamin/mineral supplement, and 4.5% consumed protein powder. Finally, 41.2% of participants were taking vitamin D supplements.

The most prevalent nutrition-related complication was distention (70.2%) (S2 Fig), whereas 70.2% of participants had bowel distention, followed by dental problems (45.1%).

As shown in Table 2, total KAP scores were not significantly related to sex, marital status, income level, and education. More than 80% of the male and female participants scored "Good" for total KAP scores. Although there was no significant relationship between income level and total KAP score, the results indicate a potential pattern in KAP score where the

**Table 1. Demographic Characteristics (n = 63).**

| Characteristic | |
|---|---|
| **Age, years** (mean±SD) | 40 ± 12.4 |
| **Presence of co-morbidities** n(%) | |
| Diabetes | 1(2) |
| Hypertension | 5(8) |
| Hyperlipidaemia/dyslipidaemia | 5(8) |
| **BMI (kg/m²)** (mean±SD) | 27.2 ± 5.6 |
| **BMI category,** n(%) | |
| Underweight (BMI <18.5) | 3(4.8) |
| Normal weight (BMI 18.5–24.9) | 17 [27] |
| Overweight (BMI 25–29.9) | 25(39.7) |
| Obese (BMI ≥ 30) | 16(25.3) |
| **Marital Status,** n(%) | |
| Single | 30(47.6) |
| Married | 24(38.1) |
| Divorced | 4(6.3) |
| Widowed | 3(4.8) |
| Refused to disclose | 2(3.2) |
| **Income Level (per month),** n(%) | |
| No income | 11(17.5) |
| >0 and ≤7000 AED | 16(25.4) |
| >7000 AED and ≤ 15000 AED | 13(20.6) |
| >15000 AED | 23(36.5) |
| **Education,** n(%) | |
| Less than high school | 15(23.8) |
| High school | 30(47.6) |
| Undergraduate education | 14(22.2) |
| Graduate degree | 4(6.3) |

*BMI: Body Mass Index; n: sample size; SD: standard deviation.

proportion of "Good" score increases with the income level. Participants with undergraduate or graduate education scored in the "Good" category. In contrast, those with less than high school or high school education had varying answers, with some falling in the "Average" category.

The majority (95.2%) of the participants in the study knew how HIV is transmitted (Table 3). However, only 12.7% knew about the side effects of the medication (ART), and 22.2% could identify high-energy-giving foods from the list. Regarding attitude and practice, most participants scored greater than 80%, except for the usage of ART on an empty stomach or two hours after food ingestion. Most of the participants (85.7%) reported consuming high-protein foods, such as meat and eggs, daily, while only 52.4% reported daily consumption of fruits and vegetables. More than 90% of the participants washed their hands frequently with soap and water (Table 3).

The overall mean (SD) of HADS-depression and HADS-anxiety scores were 4.73(3.6) and 4.90(4.9), respectively (Table 4). There were significant differences in the proportions of participants, with scores falling in the three HADS categories for both the HADS Depression and HADS-Anxiety scales (p<0.001). The study found no significant relationship between HADS depression and anxiety scores or factors such as sex, marital status, income, or education (Table 4).

## 4. Discussion

We conducted the first study (to our knowledge) that assessed knowledge, attitude, and practices (KAP) as well as the mental health status of people living with HIV (PLHIV) in the Arab Gulf Region, using a validated KAP questionnaire and HADS scale [29].The study results indicated that participants had good dietary knowledge, attitudes, and behaviors and a low prevalence of depression and anxiety.

**Table 2. Total KAP Scores Distribution among PLHIV*.**

| Total KAP score | | | | | |
|---|---|---|---|---|---|
| | | Poor (13–32) | Average (33-51) | Good (52-69) | |
| | | n(%) | n(%) | n(%) | P-value |
| **Sex** | Female | 0(0.0) | 2(15.4) | 11(84.6) | 0.630[a] |
| | Male | 1(2.0) | 4(8.2) | 44(89.8) | |
| **Marital Status** | Single | 1(3.3) | (10.0) | 26(86.7) | 0.138[b] |
| | Married | 0(0.0) | 1(4.3) | 22(95.7) | |
| | Divorced | 0(0.0) | 1(25.0) | 3(75.0) | |
| | Widowed | 0(0.0) | 1(33.3) | 2(66.7) | |
| **Income Level** | <7000 AED/month | 1(6.3) | 2(12.5) | 13(81.3) | 0.333[a] |
| | >7000 and ≤15000 AED/month | 0(0.0) | 1(7.7) | 12(92.3) | |
| | >15000 AED/month | 0(0.0) | 1(4.5) | 22(95.5) | |
| | No income | 0(0.0) | 2(18.2) | 9(81.8) | |
| **Education** | less than high school | 1(6.7) | 3(20.0) | 11(73.3) | 0.056[c] |
| | High school | 0(0.0) | 3(10.0) | 27(90.0) | |
| | Undergraduate education | 0(0.0) | 0(0.0) | 14(100.0) | |
| | Graduate education | 0(0.0) | 0(0.0) | 3(100.0) | |

The p-values correspond to Pearson's chi-square test after merging columns and rows as follows to ensure the validity of the test: (a) the columns "Poor" and "Average" were merged; (b) the columns "Poor" and "Average" were merged, and the rows "single," "Divorced", and "Widowed" were merged; (c) the columns "Poor" and "Average" were merged, and the rows "Undergraduate education" and "Graduate degree" were merged (p<0.05 is statistically significant).

The total score of all three subsections ranges between 13–69, where a score between 13 and 32 represents poor KAP values, 33–51 average, and 52–69 good values.

n: count; KAP: knowledge, attitude, & practices; PLHIV: people living with HIV.

**Table 3. Count and percentage correct responses to knowledge, attitude, and practice questions.**

| Knowledge | n(%) | Attitude | n(%) |
|---|---|---|---|
| 1. How is HIV transmitted? | 60(95.2) | 1.CD4 cell count should be regularly monitored | 55(87.3) |
| 2. What are the early symptoms of HIV/ AIDS? | 38(60.3) | 2.Important biochemical parameters should be regularly monitored | 52(82.5) |
| 3. When does Antiretroviral Treatment (ART) start? | 38(60.3) | 3. ART medicine should be consumed on an empty stomach or 2 hours after food consumption | 38(60.3) |
| 4. What are the common side effects of Antiretroviral Treatment (ART)? | 8(12.7) | 4. Sound nutrition plays an important role in the treatment of HIV/AIDS | 59(93.7) |
| 5.What is the importance of good nutrition for people diagnosed with HIV? | 43(68.3) | 5. Food intake should be altered during an episode of diarrhea | 57(90.5) |
| 6. Which are the high-energy-giving foods from the list? | 14(22.2) | 6. Well-cooked animal food should be consumed | 56(88.9) |
| 7. Which is the high protein-rich foods from the list? | 44(69.8) | 7. Personal and environmental hygiene is very important to avoid opportunistic infections | 61(96.8) |
| 8. What is the importance of fruits and vegetables in the diet of people diagnosed with HIV? | 46(73.0) | 8. Whenever needed, nutritional supplements should be prescribed by a dietitian/physician and taken | 59(93.7) |
| 9. When should oral rehydration solutions (ORS) be taken? | 27(42.9) | 9. Exercise is beneficial for patients with HIV/AIDS | 62(98.4) |
| 10. Which is a suitable exercise for HIV/AIDS patients? | 35(55.6) | 10. Not all water is safe to drink | 58(92.1) |
| 11. Which is the safest water to drink from the list? | 50(79.2) | 11. Household waste should be thrown in a covered bin | 62(98.4) |
| 12. Where should the household waste be thrown? | 48(76.2) | 12. Vegetables and fruits should be properly washed before consumption | 61(96.8) |
| 13. Where should animal food be stored? | 34(54.0) | 13. Proper hand hygiene should be practiced | 62(98.4) |
| 14. How egg should be consumed? | 47(74.6) | 14. The mouth should be properly covered while coughing and sneezing | 60(95.2) |
| **Practices** | **n(%)** | **Practices** | **n(%)** |
| 1. Do you monitor your biochemical parameters regularly? | 54(85.7) | 8. Do you consume a variety of colorful fruits daily? | 33(52.4) |
| 2. Do you monitor your weight regularly? | 46(73.0) | 9. Do you always cover your mouth while coughing and sneezing? | 56(88.9) |
| 3. Do you consume 8–10 glasses of water? | 47(74.6) | 10. Have you changed your dietary habits after your diagnosis of HIV/AIDS? | 36(57.1) |
| 4. Do you consume uncooked/raw animal-derived food items? | 44(69.8) | 11. Do you know what type of food you should consume during an episode of diarrhea? | 34(54.0) |
| 5. Do you consume milk and milk products daily? | 36(57.1) | 12. Do you wash your hands with soap frequently? | 59(93.7) |
| 6. Do you consume egg and meat products regularly? | 54(85.7) | 13. Do you exercise every day? | 12(46.0) |
| 7. Do you consume seasonal vegetables daily? | 36(57.1) | | |

The true prevalence and dynamics of HIV and nutrition-related knowledge, attitudes, and behaviors for PLHIV in the Middle East are understudied. Cultural stigma, political sensitivity, and poor healthcare infrastructure hinder PLHIV studies [35]. These obstacles impede data gathering and public health efforts [36]. The present study had a 4:1 male-to-female ratio with 20.6% female participants. This contrasts with Cameroon, which had 76.8% female participants and a 1:3 male-to-female ratio [37].

Previous research found various body weight status levels among PLHIV across countries. In the present study, 65% were either overweight or obese, while 4.8% were underweight. Banwat et al. found that 53% of North Central Nigerians were normal weight, 30% overweight, 14.4% obese, and 2.8% underweight [11]. A study conducted in Indonesia found a

**Table 4. Prevalence of anxiety and depression among the study participants.**

| | | HADS Depression | | | | HADS Anxiety | | | |
|---|---|---|---|---|---|---|---|---|---|
| | | Mean (SD) | | | | Mean (SD) | | | |
| **Total HADS Score** | | 4.73(3.6) | | | | 4.9(4.9) | | | |
| | | Normal | Borderline | Abnormal | | Normal | Borderline | Abnormal | |
| **Total HADS** | | n(%) | n(%) | n(%) | P-value | n(%) | n(%) | n(%) | P-value |
| | | 49(77.8) | 10(15.9) | 4(6.3) | <0.001[a] | 50(79.4) | 2(3.2) | 11(17.5) | <0.001[a] |
| **Sex** | Female | 11(78.6) | 3(21.4) | 0(0.0%) | 0.935[b] | 9(69.2) | 1(7.1) | 4(28.6) | 0.114[b] |
| | Male | 38(77.6) | 7(14.3) | 4(8.2) | | 41(83.7) | 1(2.0) | 7(14.3) | |
| **Marital Status** | Single | 21(70.0) | 7(23.3) | 2(6.7) | 0.307[c] | 22(73.3) | 0(0.0) | 8(26.7) | 0.498[c] |
| | Married | 21(87.5) | 2(8.3) | 1(4.2) | | 20(83.3) | 1(4.2) | 3(12.5) | |
| | Divorced | 3(75.0) | 1(25.0) | 0(0.0) | | 3(75.0) | 1(25.0) | 0(0.0) | |
| | Widowed | 2(66.7) | 0(0.0) | 1(33.3) | | 3(100.0) | 0(0.0) | 0(0.0) | |
| **Income Level (AED/month)** | <7000 | 11(68.8) | 4(25.0) | 1(6.3) | 0.125[a] | 12(75.0) | 1(6.3) | 3(18.8) | 0.233[a] |
| | 7000-15000 | 13(100.0) | 0(0.0) | 0(0.0) | | 13(100.0) | 0(0.0) | 0(0.0) | |
| | >15000 | 18(78.3) | 4(17.4) | 1(4.3) | | 17(73.9) | 1(4.3) | 5(21.7) | |
| **Educational Level** | Less than high school | 11(73.3) | 2(13.3) | 2(13.3) | 0.879[d] | 14(93.3) | 0(0.0) | 1(6.7) | 0.289[d] |
| | High school | 24(80.0) | 6(20.0) | 0(0.0) | | 22(73.3) | 2(6.7) | 6(20.0) | |
| | Undergraduate education | 10(71.4) | 2(14.3) | 2(14.3) | | 10(71.4) | 0(0.0) | 4(28.6) | |
| | Graduate education | 4(100.0) | 0(0.0) | 0(0.0) | | 4(100.0) | 0(0.0) | 0(0.0) | |

HADS = hospital anxiety & depression score, n = count. For both HADS-Anxiety & HADS-Depression scoring system: normal (0–7), borderline (8–10), mild, high (11–21) clinical case. (p < 0.05 is statistically significant); AED = UAE Dirhams

(a)The p-value is based on the chi-square test for goodness-of-fit, where the null hypothesis is equal proportion across the three HADS categories (Normal, Borderline, Abnormal).

(b)The p-value is based on Pearson's chi-square test after merging the columns "Borderline" and "Abnormal" to ensure the validity of the test.

(c)The p-value is based on Pearson's chi-square test after merging the columns "Borderline" and "Abnormal" and the rows "Divorced" and "Widowed" to ensure the validity of the test.

(d)The p-value is based on Pearson's chi-square test after merging the columns "Borderline" and "Abnormal" and the rows "Undergraduate education" and "Graduate education."

mean BMI of 20.97 kg/m², ranging from 14 to 28 kg/m² [12]. Perpetue et al. (2021) found 8.5% of the survey participants in Cameroon were underweight.

The present study found that 17% of participants took oral nutritional supplements composed of a balanced nutritional composition of protein, lipids, and carbohydrates. Lipid-based dietary supplements can boost weight gain, lean body mass, and grip strength in HIV patients commencing ART while whey-containing supplements improve immunological recovery [38].

Nutrition knowledge improves PLHIV diets by increasing the number of meals and frequency of nutrient-rich foods such as fruits, vegetables, legumes, animal products, and grains. These foods could improve immunity, minimize muscle loss, and the overall nutritional status. Larasati et al. (2019) found a significant relationship between knowledge and energy intake but no significant relationship between attitude and food intake [12]. Our study found that 87.3% of participants demonstrated good KAP with no significant difference between males and females. In contrast, a study conducted in India found that 52% of the participants had low knowledge, 44% had average, and only 2% had good knowledge [28].

In research on depression and anxiety, PLHIV had significantly higher scores in depression and anxiety compared to the general population [39]. The prevalence of HADS depression and anxiety scores of the participants in the present study were 8.0% and 17.5%, respectively. This finding is in contrast to previous research indicating a higher prevalence

of depression and anxiety among PLHIV [40]. A study conducted in Pakistan found that 89.9% had depression and 80.3% had anxiety [39].

The present study found bowel distention as the most common nutrition-related complication, which was reported by 70% of the participants. A previous study indicated that HIV patients face nutritional concerns, including diarrhea (36%), poor appetite (34%), and nausea (26%) [41].

### 4.1 Strengths & limitations

The current study has a number of noteworthy strenghts. To the best of the authors' knowledge, it is the first study to examine knowledge, attitudes, practices, and mental health among individuals living with HIV in the Arabian Gulf Region and one of the few in the larger Middle East. The study's methodological rigor is further improved by the use of validated and standardized data collection tools. Moreover, despite recruitment challenges due social stigma, privacy concerns, and low awareness, the sample included 36% of all HIV patients who visited the outpatient clinic. The study's main limitations include the sample recruited from a single clinic and thus may not be representative of PLHIV in the UAE. However, to minimize any potential bias, the participants were recruited sequentially and validated tools were used for the data collection.

### 5. Conclusion

The study's participants showed positive attitudes and knowledge about nutrition, and the prevalence of anxiety and depression was less than 20%, which is significantly lower than estimates for PLHIV worldwide. According to international data, anxiety affects about 22% of PLHIV and depression affects 28–34% [42,43], highlighting the relatively lower mental health burden seen in this study's population. This lower mental health burden of the study participants could be the result of easily accessible HIV services, or the robust support for ART adherence by healthcare team.

This study offers crucial baseline data and emphasizes the need to further incorporate nutrition education and mental-health support into HIV care programs. It is the first study to evaluate nutrition-related knowledge, attitudes, and practices among PLHIV in the Arabian Peninsula and one of the few in the Middle East to concurrently examine mental health.

### 5.1 Implications for research practice

This study has several implications for research practice. First, successful recruitment of a highly stigmatized and rare population underscores the importance of confidentiality, close collaboration with the relevant clinic staff, and a rapport-building, culturally sensitive approach to care of PLHIV. Such approaches may be highly beneficial when trying to reach similar populations in this region. Secondly, the participants in this study demonstrated a generally sound nutritional knowledge, positive attitudes, and good practices. Further research is warranted to explain whether this is due to advice given at clinics, support from friends and family, or a robust healthcare system in general. Understanding how these positive outcomes are achieved could guide practitioners in designing and implementing strategies to strengthen nutrition knowledge, attitudes, and practices among populations that do not perform as well.Moreover, our findings indicate that future research should consider integrating nutrition education programs with mental health support, since living with HIV can be highly stressful. Finally, multicenter studies in the MENA region is warranted to generate stronger evidence for culturally appropriate healthcare services for PLHIV that can improve adherence and reduce HIV transmission.

### Supporting information

**S1 Fig. Types of nutritional supplements used by the participants.**
(TIFF)

**S2 Fig. Prevalence of nutrition-related complicatio.**
(TIFF)

## Acknowledgments

The authors would like to thank the participants of this study. The authors are grateful to the Infectious Disease Unit at Rashid Hospital in the Emirate of Dubai, UAE, for facilitating this research.

**AI Use Statement**: AI tools were limited to language editing. The authors created and verified all scientific content.

## Author contributions

**Conceptualization:** Souheir M. Alia, Habiba I. Ali.

**Data curation:** Souheir M. Alia.

**Formal analysis:** Souheir M. Alia, Taoufik L. Zoubeidi.

**Investigation:** Souheir M. Alia.

**Methodology:** Souheir M. Alia, Taoufik L. Zoubeidi, Habiba I. Ali.

**Project administration:** Souheir M. Alia.

**Supervision:** Habiba I. Ali.

**Validation:** Taoufik L. Zoubeidi, Habiba I. Ali.

**Visualization:** Souheir M. Alia.

**Writing – original draft:** Souheir M. Alia, Taoufik L. Zoubeidi, Habiba I. Ali.

**Writing – review & editing:** Taoufik L. Zoubeidi, Habiba I. Ali.

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
