## [Decision Letter · Decision Letter 0]

14 Nov 2025

Dear Dr. Ali,

Thank you for submitting your manuscript to PLOS ONE. After careful consideration, we feel that it has merit but does not fully meet PLOS ONE’s publication criteria as it currently stands. Therefore, we invite you to submit a revised version of the manuscript that addresses the points raised during the review process.

The reviewers have both noted that there are some areas of the manuscript that can be improved to enhance clarity this includes some grammatical type corrects as well as improving the Discussion.

We look forward to receiving your revised manuscript.

Kind regards,

Jenny Wilkinson, PhD

Academic Editor

PLOS ONE

Journal Requirements:

2. Please remove all personal information, ensure that the data shared are in accordance with participant consent, and re-upload a fully anonymized data set.

Reviewers' comments:

Reviewer's Responses to Questions

**Comments to the Author**

1. Is the manuscript technically sound, and do the data support the conclusions?

Reviewer #1: Yes

Reviewer #2: Partly

2. Has the statistical analysis been performed appropriately and rigorously?

Reviewer #1: Yes

Reviewer #2: I Don't Know

3. Have the authors made all data underlying the findings in their manuscript fully available?

Reviewer #1: Yes

Reviewer #2: Yes

4. Is the manuscript presented in an intelligible fashion and written in standard English?

Reviewer #1: Yes

Reviewer #2: Yes

Reviewer #1: Thank you for the opportunity to review this manuscript. This study provides interesting findings on the knowledge, attitudes, and practices of those living with HIV in the middle east. I think it is well-written, although it could use some work on improving the discussion and key takeaways.

Below are comments to the authors to improve the paper:

Abstract: In the conclusions, you can delete “knowledge” before KAP

Introduction: I recommend a transition sentence between the 3rd and 4th paragraphs – provide a segue from HIV prevalence to health behaviors, which naturally leads to discussing nutrition knowledge and behaviors. I also recommend re-working the introduction section a little bit as it sort of jumps back and forth between HIV and nutrition – perhaps start with the macro issue (HIV is problematic and mention its prevalence), then discuss how it leads to mental health issues and poor attitudes, then mention how it leads to poor behaviors which are understudied in MENA.

Introduction, line 24: you might define BMI and ART here for readers unaware

Section 2.3, line 70: You can delete “(KAP)” in this line

Section 2.4: Perhaps add some brief information on the final questionnaire instrument such as total number of questions and average time to complete it

Section 2.5, line 106: Not sure what is meant by “[23] guided all operations.” Can you clarify?

Section 2.6: How was missing data handled? Please include that here

Lines 126 and 127 have repeated information that can be deleted or the sentences combined into one

Line 219: Do you mean “Cameroon”?

Lines 220-225: Not sure what workplace prejudice has to do with diet composition. I recommend changing the theme sentence of this paragraph

Implications for practice section: The first two sentences in this section mostly just repeat information already provided and can be deleted, then this section revised. One thing missing in this section, and the paper overall, is a key takeaway – what does one do with the information from this study? Do the results inform future interventions? Was there a recruitment strategy that was particularly helpful for this hard to reach population? Given generally “good” levels of KAP among these participants, what can or should be done, if anything? Is there a need to find out why their KAP was so good and disseminate strategies to other populations to improve their KAP? Please consider discussing what is/are the important things the reader can derive from the study

Reviewer #2: 1. References 1 and 2 reference is same

2. As of 2023, roughly 1,700 residents in the United Arab Emirates (UAE) had HIV, giving the overall population a prevalence rate of 0.02% [4]. (cite proper reference, doesn’t correspond, latest data from WHO country profile is of 2020.) 2023 data with residents’ number provided in this line needs clarification. More details required in this paragraph, local outlook on the matter.

3. Line 25 - correlated with good KAP (Banwat, 2016). Not referenced, while banwat reference is present as no 27 ?

4. Line 27 – reference [7] stated about another study but not reference correct

5. Line 30 - patients often experience depression (HIV/AIDS and Mental Health, 2022).Citation style differing

6. The nutrition-related KAP of HIV/AIDS patients in the MENA region has not been previously

examined despite their heightened vulnerability to malnutrition. Moreover, no studies on PLHIV in the Arabian Middle East Region exist. This Statement need to thought through : as there are studies :

1. https://journals.plos.org/plosone/article?id=10.1371/journal.pone.0288838

2. https://pmc.ncbi.nlm.nih.gov/articles/PMC10001308/

3. https://www.frontiersin.org/journals/nutrition/articles/10.3389/fnut.2024.1294233/full

7. Line 49-50 states nutrition-related KAP and mental health status (depression and anxiety), Why was depression and anxiety taken, rationale and prevalence of the same.

8. Line 51 - in an interventional study. What interventional study ?? related??

9. Line 106 -exclusive access to the data to protect confidentiality. [23] guided all operations. Rephrase

10. Supplementary file looks like a chatgpt AI image (with background grids)

11. Demographic details – did u assess nationality ?? or region ?? or occupation ?

12. Table 3 – under knowledge 11. Which is the safest water to drink from the list?. What list ? Table 3 need refinement

13. Line 203 - (to our knowledge), First study is enough, no requirement. If mentioning please reframe

14. Line 205- HADS scale (Zigmond & Snaith , Reference style check

15. Line 254-255 - sample corresponds to 36% of the total population of HIV patients , how can that be a strength please explain, and as it is cross sectional design – Mention the biases linked to your study, also comment on generalisability. How did you manage to overcome bias? Please mention all limitation in one subheading

16. Conclusion needs more detailing, Abstract mention prevalence of mental condition less than 20% , why quote less than 20% , Significance of the number.

17. Lastly references need through checking, some places its APA style, some references repeated, some wrongly cited.

**Do you want your identity to be public for this peer review?** For information about this choice, including consent withdrawal, please see our Privacy Policy

Reviewer #1: No

Reviewer #2: **Yes:** Rifah Anwar Assadi

---

## [Author Response · Author response to Decision Letter 1]

31 Jan 2026

PONE-D-25-26744

Nutrition-related knowledge, attitudes, practice, and mental health status of people living with HIV in Dubai, United Arab Emirates: A cross-sectional descriptive study

PLOS ONE

Response to Reviewers’ Comments

We thank the reviewers for their constructive comments. We have addressed all these comments in the revised manuscript, point-by-point, as described below.

Reviewer 1:

Thank you for the opportunity to review this manuscript. This study provides interesting findings on the knowledge, attitudes, and practices of those living with HIV in the middle east. I think it is well-written, although it could use some work on improving the discussion and key takeaways. Below are comments to the authors to improve the paper:

Reviewer: Abstract: In the conclusions, you can delete “knowledge” before KAP

Response: Thank you for pointing it out. It has been removed.

Reviewer: Introduction: I recommend a transition sentence between the 3rd and 4th paragraphs – provide a segue from HIV prevalence to health behaviors, which naturally leads to discussing nutrition knowledge and behaviors. I also recommend re-working the introduction section a little bit as it sort of jumps back and forth between HIV and nutrition – perhaps start with the macro issue (HIV is problematic and mention its prevalence), then discuss how it leads to mental health issues and poor attitudes, then mention how it leads to poor behaviors which are understudied in MENA.

Response: Thank you for making these important points. We have added a transition sentence. We have added in the third paragraph about the prevalence of HIV in the UAE and elaborated on it. Moreover, we have re-arranged the introduction in which as you have kindly suggested that I start with living with HIV and its implications on the mental health status and how that affects healthy versus risky behaviours which in turn has a direct effect on the overall prevalence of HIV in the MENA region and more specifically the UAE. The changes are in the 3rd, 4th, and 5th paragraphs under the introduction – highlighted in yellow.

Reviewer: Introduction, line 24: you might define BMI and ART here for readers unaware

Response: Thank you for this remark, it is done and highlighted (please refer to the 5th paragraph of the introduction).

Reviewer: Section 2.3, line 70: You can delete “(KAP)” in this line

Response: Thank you, it has been deleted and highlighted.

Reviewer: Section 2.4: Perhaps add some brief information on the final questionnaire instrument such as the total number of questions and average time to complete it

Response: Thank you, good point. It has been added to section 2.4 and highlighted.

Reviewer: Section 2.5, line 106: Not sure what is meant by “[23] guided all operations.” Can you clarify?

Response: Thank you for bringing this to our attention; edited, please refer to the revised section 2.5.

Reviewer: Section 2.6: How was missing data handled? Please include that here

Lines 126 and 127 have repeated information that can be deleted or the sentences combined into one

Response: Thank you for this comment. We have added two statements highlighted in yellow in sections 2.3 and 2.6 on how missing data was handled. Moreover, we have deleted the repeated information in the results section describing Figure 2 (Supplementary material).

Reviewer: Line 219: Do you mean “Cameroon”?

Response: Thank you, corrected.

Reviewer: Lines 220-225: Not sure what workplace prejudice has to do with diet composition. I recommend changing the theme sentence of this paragraph

Implications for practice section: The first two sentences in this section mostly just repeat information already provided and can be deleted, then this section revised. One thing missing in this section, and the paper overall, is a key takeaway – what does one do with the information from this study? Do the results inform future interventions? Was there a recruitment strategy that was particularly helpful for this hard to reach population? Given generally “good” levels of KAP among these participants, what can or should be done, if anything? Is there a need to find out why their KAP was so good and disseminate strategies to other populations to improve their KAP? Please consider discussing what is/are the important things the reader can derive from the study

Response: Thank you for highlighting this, we have removed the prejudice part in comparison to diet composition. It is highlighted in the discussion section. As to the second point relating to the implications of this research, first, thank you for this important suggestion. The paragraph has been rewritten.

Reviewer 2:

Reviewer: 1. References 1 and 2 reference is same

Response: Thank you for pointing it out. It has been fixed and highlighted in yellow in the first paragraph of the introduction as well as in the references section.

Reviewer: 2. As of 2023, roughly 1,700 residents in the United Arab Emirates (UAE) had HIV, giving the overall population a prevalence rate of 0.02% [4]. (cite proper reference, doesn’t correspond, latest data from WHO country profile is of 2020. 2023 data with residents’ number provided in this line needs clarification. More details required in this paragraph, local outlook on the matter.

Response: Thank you for pointing it out. It has been fixed and re-written, re-referenced and highlighted in yellow (please refer to the third paragraph under introduction).

Reviewer: 3. Line 25 - correlated with good KAP (Banwat, 2016). Not referenced, while banwat reference is present as no 27?

Response: Thank you for noticing that, fixed and highlighted in yellow. All references were revised; Banwat, 2016, is now reference number 11.

Reviewer: 4. Line 27 – reference [7] stated about another study but not reference correct

Response: Referenced correctly and highlighted in yellow.

Reviewer: 5. Line 30 - patients often experience depression (HIV/AIDS and Mental Health, 2022).Citation style differing

Response: Thank you for noticing that. Fixed and highlighted; it is reference number 6.

Reviewer: 6. The nutrition-related KAP of HIV/AIDS patients in the MENA region has not been previously examined despite their heightened vulnerability to malnutrition. Moreover, no studies on PLHIV in the Arabian Middle East Region exist. This Statement need to thought through : as there are studies :

1. https://journals.plos.org/plosone/article?id=10.1371/journal.pone.0288838

2. https://pmc.ncbi.nlm.nih.gov/articles/PMC10001308/

3. https://www.frontiersin.org/journals/nutrition/articles/10.3389/fnut.2024.1294233/full

Response: Thank you for listing these interesting studies, I agree with you. However, there were not studies addressing KAP of PLHIV. The first study is KAP of health care professionals in the MENA region, the second one is basically the prevalence of HIV in MENA region, and the third one is in Ethiopia which is in East Africa.

Reviewer: 7. Line 49-50 states nutrition-related KAP and mental health status (depression and anxiety), Why was depression and anxiety taken, rationale, and prevalence of the same.

Response: Thank you for raising this point. It was a secondary outcome measured and the main primary outcome of the study was KAP score of PLHIV in the UAE. The rationale and prevalence are mentioned in the last paragraph of the introduction and highlighted in yellow.

Reviewer: 8. Line 51 - in an interventional study. What interventional study ?? related??

Response: We apologize for the confusion and thank you for pointing this out. Kindly find the type of study under section 2.1 of the methods section; the first line states the type of study conducted.

Reviewer: 9. Line 106 -exclusive access to the data to protect confidentiality. [23] guided all operations. Rephrase

Response: Thank you for pointing this out; it has been done and highlighted under Ethical Approval (2.5) in the methods section.

Reviewer: 10. Supplementary file looks like a chatgpt AI image (with background grids)

Response: Thanks for your valuable comment. This was a picture of a graph generated from SPSS version 28.0; however, I have re-done the figures on Excel, saved in .tiff files and with 300 dpi resolution.

Reviewer: 11. Demographic details – did u assess nationality ?? or region ?? or occupation ?

Response: Thank you for seeking this clarification. All the patients were Arabs – UAE nationals. Participant occupation was not assessed; however, their income level was (Table 1). However, I added a statement in the first line under results stating that they were all UAE nationals and highlighted it in yellow. All participants who attended the outpatient clinic were UAE nationals.

Reviewer: 12. Table 3 – under knowledge 11. Which is the safest water to drink from the list?. What list ? Table 3 need refinement

Response: Thank you for your feedback. The table listed all the questions from the KAP questionnaire as is, without the potential answers. Hence, the exact wording of each question was included in the table, covering the questions asked in the knowledge, attitude, and practices subsections of the KAP questionnaire

Reviewer: 13. Line 203 - (to our knowledge), First study is enough, no requirement. If mentioning please reframe

Response: Thank you; fixed. “This study is the first,…” in the conclusion of the abstract.

Reviewer: 14. Line 205- HADS scale (Zigmond & Snaith , Reference style check

Response: Thank you for pointing it out, it is has now been corrected to the proper referencing style. It is number 29in the references section as well as within text (fiirst paragraph of discussion).

Reviewer: 15. Line 254-255 - sample corresponds to 36% of the total population of HIV patients , how can that be a strength please explain, and as it is cross-sectional design – Mention the biases linked to your study, also comment on generalisability. How did you manage to overcome bias? Please mention all limitation in one subheading

Response: Thank you for this valuable comment. We have updated the Discussion section in Section 4.1.

Reviewer: 16. Conclusion needs more detailing, Abstract mention prevalence of mental condition less than 20% , why quote less than 20% , Significance of the number.

Response: Thank you for seeking clarification. It has been expanded, and your comments were addressed and highlighted in both the conclusion section under the discussion section as well as the abstract (conclusion).

Reviewer: 17. Lastly references need through checking, some places its APA style, some references repeated, some wrongly cited.

Response: Thank you for emphasizing this issue. The in-text citations and the References have been reviewed and fixed throughout the manuscript.

---

## [Editor Report · Decision Letter 1]

18 Feb 2026

Nutrition-related knowledge, attitudes, practice, and mental health status of people living with HIV in Dubai, United Arab Emirates: A cross-sectional descriptive study

PONE-D-25-26744R1

Dear Dr. Ali,

We’re pleased to inform you that your manuscript has been judged scientifically suitable for publication and will be formally accepted for publication once it meets all outstanding technical requirements.

Kind regards,

Jenny Wilkinson, PhD

Academic Editor

PLOS One

Additional Editor Comments (optional):

Thank you for your revisions, these have satisfactorily addressed reviewer comments.
---

## [Editor Report · Acceptance letter]

PONE-D-25-26744R1

PLOS One

Dear Dr. Ali,

I'm pleased to inform you that your manuscript has been deemed suitable for publication in PLOS One. Congratulations! Your manuscript is now being handed over to our production team.

Kind regards,

on behalf of

Dr Jenny Wilkinson

Academic Editor

PLOS One